# Interpretable and Adaptive Graph Contrastive Learning with Information Sharing for Biomedical Link Prediction

## Abstract

The identification of unobserved links in drug-related biomedical networks is essential for various drug discovery applications, which is also beneficial for both disease diagnosis and treatment through exploring the underlying molecular mechanisms. However, existing solutions face significant challenges due to three main limitations: (1) lack of interpretability to provide comprehensive and reliable insights, (2) insufficient robustness and flexibility in cold-start scenarios, and (3) inadequate interaction and sharing of multi-view information. In light of this, we propose DrugXAS, an interpretable and adaptive cross-view contrastive learning framework with information sharing for biomedical link prediction. Specifically, DrugXAS has three distinctive characteristics for addressing these challenges. To solve the first problem, we propose an attention-aware augmentation scheme to provide understandable explanations of intrinsic mechanisms. To deal with the second challenge, we propose an adaptive graph updater and neighborhood sampler, which select proper neighbors according to the feedbacks from the model to improve aggregation ability. To tackle the third issue, an information sharing module with diffusion loss is proposed to incorporate chemical structures into heterogeneous relational semantics and facilitate the contrast process. Empirically, extensive experiments on seven benchmark datasets involving multi-type tasks demonstrate that the proposed DrugXAS outperforms the state-of-the-art methods in terms of precision, robustness, and interpretability. The source code of DrugXAS is available at https://anonymous.4open.science/r/DrugXAS-8EC7.

## 1 Introduction

Discovering unknown molecular interactions and associations within biomedical networks is immensely valuable in practical applications (Muzio et al., 2020), including finding novel biomarkers for diseases, repurposing drugs for other purposes, and identifying drug side effects (Ding et al., 2024a). While a vast number of links have been identified through wet experiments or clinic reports, many remain undiscovered (Zhong & Mottin, 2023). Furthermore, determining potential links using wet experimental techniques is extremely expensive, time-consuming, and labor-intensive (Feng et al., 2024b). Recently, researchers have exploited computational methods to predict links in drug-related biomedical networks and achieved remarkable success (Ma et al., 2023; Li et al., 2022). Considering that interactions/associations between molecules and entities can be readily expressed as networks, with molecules as nodes and interactions as edges, graph neural networks (GNNs) have demonstrated extraordinary capabilities and potential in biomedical link prediction tasks (Wang et al., 2024a; Liu et al., 2024b). Despite numerous efforts and impressive achievements, the current research state of biomedical link prediction faces several key challenges that limit the application. Here, we highlight the following three issues and deficiencies that need to be addressed:

(1) **Interpretability.** Interpretability, the capacity to explain and elucidate the outputs of deep learning models, is particularly significant in drug-related research since it has a tremendous impact on patients' trust and assists medical professionals (Ding et al., 2024b). There is an emerging demand for drug discovery methods that offer greater reliability and aid in the comprehension of underlying models. However, a significant obstacle of existing drug-related link prediction lies in their lack

of interpretability (Feng et al., 2024a; Yu et al., 2021). To this end, there is still ample scope for investigation in the field of advanced interpretability to produce explanations that are more comprehensive and flexible, which is beneficial for understanding the inherent mechanisms and patterns of the task. In essence, this leads us to the following problem: *how to provide reasonable and meaningful explanations for predictions to achieve interpretability from a biomedical perspective?*

(2) **Cold-Start Scenario Applications.** Existing models mainly focus on datasets where drugs and entities are randomly split, with drugs and entities in the test set having already appeared in the training set. Nonetheless, this situation is far from realistic, where the majority of drugs or entities are absent from the training set (Pahikkala et al., 2014), leading to a dramatic drop in prediction performance of current methods (Wang et al., 2022). This issue, refers to the cold-start problem in recommendation systems, severely restricts the practical

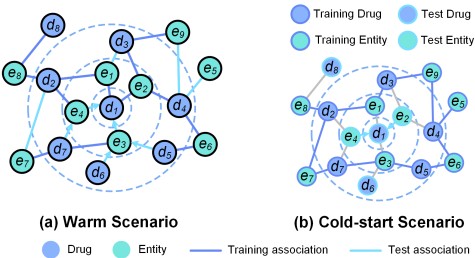

Figure 1: The illustration of cold-start problem.

application of biomedical link prediction techniques (Ye et al., 2021). Learning robust and stable representations of novel drugs and entities in the absence of known correlation information is one of the primary challenges facing cold-start link prediction (Hao et al., 2021). For instance, in the warm scenario depicted in Figure 1(a), messages from high-order neighbors can be aggregated to target node drug $d_1$. However, in the cold-start scenario, only the 1-order neighbor entities $e_2$ and $e_4$ can be utilized to represent $d_1$, which are both cold-start nodes without known associations, leading to insufficient information and suboptimal representations. Towards this end, the cold-start problem motivates us the second research question: *how to enhance the representation learning capability for cold-start scenarios?*

(3) **Exploration of Multi-view Information.** Although graph contrastive learning (GCL), particularly cross-view GCL, has emerged as an effective representation learning framework for biomedical link prediction (Zhang et al., 2024), current GCL-based models only consider a portion of biomedical knowledge (Ma et al., 2023). Since molecular chemical information and biomedical interaction modeling frequently have heterogeneous rather than monomorphic dependencies, with the former focusing on fine-level and the latter based on coarse-level, it is crucial to achieve interaction and sharing of information between the two views (Xie et al., 2024; Wu et al., 2024). Each drug has unique chemical traits and a variety of link patterns for the inherent interaction mechanisms, thus the influence and contribution of different source of data may be entirely different. For instance, molecular structure information is more vital for tasks in which molecular actions and mechanisms play a key role, such as drug-target interactions (DTIs) and drug-side effects (DSEs). Blindly encoding features without considering their individual and distinctive characteristics easily results in suboptimal learned embeddings (Liu et al., 2024a). In view of this obstacle, we are intuitively inspired to ask the following question: *how to effectively interact and share knowledge across different views to make full use of both link and auxiliary information?*

To address the above-mentioned challenges in **Drug**-related biomedical link prediction, a novel Interpretable (**X**) and **A**daptive contrastive learning model with Information **S**haring is proposed, termed as DrugXAS. We highlight our primary contributions as follows: (1) *To overcome the first limitation*, DrugXAS devises a novel attention-aware graph augmentation strategy, enabling our model adaptively focus on important molecular structures and potential biomedical associations. Moreover, the network-view augmentation facilitates DrugXAS in providing rational explanations of the prediction results thereby achieving interpretability. (2) *To tackle the second challenge*, DrugXAS proposes an adaptive graph updater with a neighborhood sampler to dynamically select effective neighbors according to the feedbacks from the encoder. (3) *To handle the third issue*, DrugXAS proposes an information sharing module between molecule- and network-view components (InSMN) and a diffusion loss scheme to further advance cross-view contrastive learning. The information sharing module extracts topological and chemical information from two views simultaneously, while the diffusion loss aims to bring the representations of the two views closer, optimizing contrastive learning. (4) Extensive experiments are conducted on seven public datasets of different types of tasks, demonstrating the superiority of DrugXAS compared with state-of-the-art methods under both warm and cold-start scenarios.

## 2 RELATED WORK

**Biomedical Link Prediction.** The goal of drug-related biomedical link prediction is to predict whether there is a link between a particular pair of drugs and entities (Wang et al., 2024b). In particular, SkipGNN (Huang et al., 2020) introduces skip similarity into GNNs by aggregating information from low to high orders. SiGrac (Coşkun & Koyutürk, 2021) designs a novel node similarity-based graph convolution process with multiple measures of similarity. Zhao et al. (2021) propose CSGNN, which leverages the GNN aggregator from mix-hop neighborhoods and utilizes contrastive learning for multi-task prediction. MVGCN (Fu et al., 2021) devises a multi-view graph convolution network (GCN) with inter- and intra-view neighborhood aggregators. HOGCN (KC et al., 2022) focuses on high-order information, aggregating neighbor nodes information at different distances. Shen et al. (2024) propose CGCN, which adopts a curvature-enhanced GCN to learn local geometric attributes. Li et al. (2024a) propose SubKNet, which employs a graph kernel neural network to the extracted subgraphs for learning node representations.

**Graph Contrastive Learning.** Contrastive learning aims to maximize the agreement between positive samples while pushing away negative samples under appropriate data augmentations (Li et al., 2024b). Here we primarily focus on GCL methodologies. For example, Veličković et al. (2018) propose DGI, which creates positive samples by combining local and global semantics while the negatives are represented by nodes in a randomly damaged network. GMI (Peng et al., 2020) proposes a fine-grained contrastive loss by comparing the topological structure and node properties. Hassani & Khasahmadi (2020) propose MVGRL, which exploits cross-view contrastive learning of nodes and graphs. With respect to heterogeneous graph representation learning, DMGI (Park et al., 2020) performs contrastive learning between the original network and the corrupted network on each individual view and meta-path. Wang et al. (2021) propose HeCo, which devises two views of the heterogeneous graph, learning node features from both local and high-order structures. H-GCL (Zhu et al., 2023) is proposed to employ hypergraph to establish the augmented view for learning high-quality embeddings.

## 3 PRELIMINARY

**Biomedical Heterogeneous Graph.** A biomedical heterogeneous graph can be defined as an undirected graph $\mathcal{G} = (\mathcal{V}, \mathcal{E})$, with a node type mapping function $\phi : \mathcal{V} \to \mathcal{O}$ and an edge type mapping function $\varphi : \mathcal{E} \to R$. $\mathcal{V}$ represents the set of nodes which corresponds to drugs and other types of biomedical entities such as targets, diseases and side effects, etc. $\mathcal{E}$ is the set of edges (i.e., links) between nodes in $\mathcal{V}$, corresponding to associations or interactions between biomedical entities.

**Drug Molecular Graph.** Given a drug $d_i$, the molecular graph is formulated as $\mathcal{G}_{d_i} = (\mathcal{M}, \mathcal{A})$, where $\mathcal{M}$ denotes the set of atoms of the drug and $\mathcal{A}$ represents the set of bonds between atoms.

**Biomedical Link Prediction.** Given the drug-entity pair $(d_i, e_j)$ in biomedical graph $\mathcal{G}$, our work aims to evaluate the probability score between this pair of drug and entity with correlation heterogeneity, i.e., learn a mapping function to predict the link probability between drug $d_i$ and entity $e_j$.

**Cold-start Scenario.** (1) Cold-drug Task: Each drug that appears in the training set does not appear in the test set, while each entity can appear in both training and test set. (2) Cold-entity Task: Each entity that appears in the training set does not appear in the test set, while each drug can appear in both training and test set. (3) Cold-pair Task: Neither the drug nor the entity in the training set appear in the test set.

## 4 METHOD

In this section, we elaborate the detailed architecture of our proposed DrugXAS, an interpretable and adaptive cross-view contrastive learning model with information sharing module, whose overall framework is sketched in Figure 2.

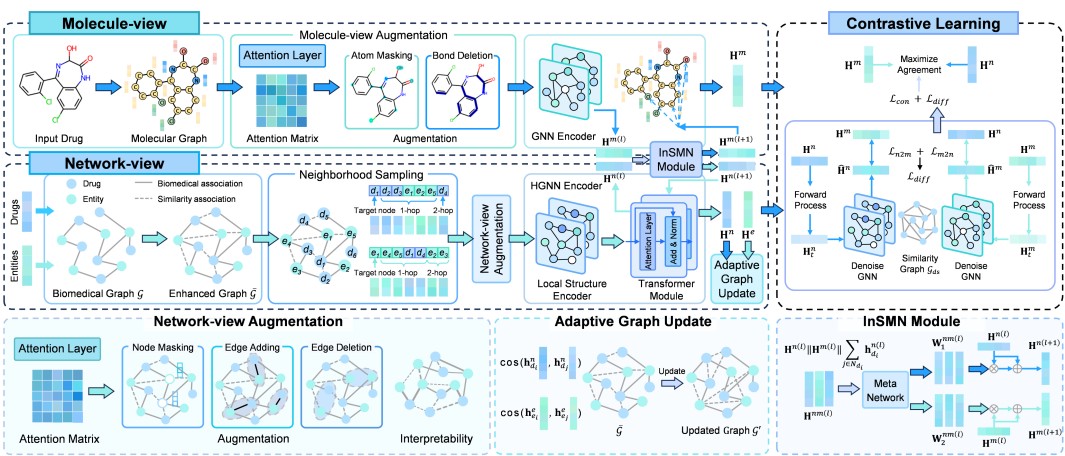

Figure 2: The overall framework of our proposed DrugXAS.

## 4.1 MOLECULE-VIEW REPRESENTATION LEARNING

**Drug Molecular Graph Construction.** To obtain chemical property as auxiliary information, drugs are modeled as undirected molecular graphs to learn representations from molecule-view encoder. For the given drug $d_i$, we construct the corresponding molecular graph $G_{d_i}$ with 41-dimensional node features (Jiang et al., 2021).

**Molecule-view Augmentation.** Inspired by (Chen et al., 2023b; Liu et al., 2022), we design an attention-aware augmentation strategy for generating high-quality augmented molecular graphs, which serves as a selection index to measure the contribution and importance of each atom and bond. We propose to use the attention mechanism of GATv2 (Brody et al., 2021) to learn the information strength of atoms and bonds, as follows:

$$\alpha_{a_i,a_j} = \frac{exp\left(\mathbf{a}^T\left(LeakyReLU\left(\mathbf{W}\left[\mathbf{h}_{a_i}\|\mathbf{h}_{a_j}\right]\right)\right)\right)}{\sum_{k\in N_{a_i}} exp\left(\mathbf{a}^T\left(LeakyReLU\left(\mathbf{W}\left[\mathbf{h}_{a_i}\|\mathbf{h}_{a_k}\right]\right)\right)\right)}. \tag{1}$$

where $\alpha_{a_i,a_j}$ denotes the calculated attention score between atom node $a_i$ and $a_j$. Subsequently, we employ two molecular graph augmentation methods for input drugs according to the learned attention matrix, i.e., atom masking and bond deletion, whose detailed implementation is presented in Appendix A due to the page limitation. Instead of utilizing a fixed augmentation ratio, we employ a variable and dynamic augmentation ratio that allows our model to sustain stable performance regardless of fluctuations in the amount of input information (Tian et al., 2023). To be concrete, the augmentation ratio is gradually increased with regard to the training epoch, following a linear change that enables the model to adaptively learn from easier to more difficult scenarios. Mathematically, the augmentation ratio at epoch $t_e$ can be defined as $p_{t_e} = p_0 + t_e \cdot \Delta p$, where $p_0$ is the initial ratio and the maximum ratio is set to $p_x$.

**GNN Encoder.** The augmented molecular graphs are then put into GNN encoders with the mean pooling readout function. Here we also apply GATv2 as the backbone encoder due to its universal and powerful attention layer. We denote the encoded molecule-level embedding as $\mathbf{H}^m \in \mathbb{R}^{n_d \times d}$.

## 4.2 NETWORK-VIEW REPRESENTATION LEARNING

**Biomedical Heterogeneous Graph Construction.** In the network-view, for better adaptation to cold-start scenarios, we first supplement the constructed biomedical heterogeneous graph $\mathcal{G}$ with additional similarity information. To be specific, we select the top $k_n$ nodes with the highest similarity for each node and add them as new edges to construct the enhanced graph $\bar{\mathcal{G}}$.

**Dynamic Neighborhood Sampling.** Drawing inspiration from previous work (Mao et al., 2023), we sample the fixed number of neighbors of each node as neighborhood sequence for message

propagation. Given the target node, we iteratively add its neighbors from 1-hop to higher hop to the neighborhood sequence until the sequence length reaches $s_n$. Formally, taking target drug $d$ as an example, the sampled neighborhood can be represented as $S_d = [d, d_1, \ldots, d_{s_n-1}]^\mathrm{T}$, and the corresponding feature embedding is denoted as $\mathbf{H}_d^S = \left[\mathbf{h}_d, \mathbf{h}_{d_1}, \ldots, \mathbf{h}_{d_{s_n-1}}\right]^\mathrm{T} \in \mathbb{R}^{s_n \times d}$. The fixed and unified length of neighborhood facilitates the capture of long-range dependencies, and is compatible with the subsequent attention layer and HGNN encoder. Furthermore, the neighborhood sampling operation is implemented at every epoch, resulting in dynamically sampled neighborhoods that introduce stochastic uncertainty to enhance the generalization of our model.

**Network-view Augmentation.** The network-view augmentation is also based on the learned attention matrix. To be precise, the attention layer of Eq.(1) is applied on each neighborhood sequence to produce the attention matrix as a guidance of data augmentation. The three types of augmentation strategy employed in the network-view are illustrated in Appendix A, including node masking, edge adding, and edge deletion. The augmentation ratio in this view is also dynamically variable.

**HGNN Encoder.** Local structure information has been proven to be essential for the HGNN model (Lv et al., 2021). We introduce GCN (Kipf & Welling, 2016) to provide local sub-structure enhanced node features, which is simple yet competitive for HGNN since that the heterogeneity can be learned through heterogeneous feature projection (Liu et al., 2023). Then we employ the transformer encoder (Vaswani et al., 2017) to capture information from long-range dependencies. The entire inputs of the transformer module for drugs and entities are the sampled neighborhood sequences, denoted by $\mathbf{H}^{ns(0)} \in \mathbb{R}^{n_d \times s_n \times d}$ and $\mathbf{H}^{es(0)} \in \mathbb{R}^{n_e \times s_n \times d}$, respectively. We modify the two main components in transformer to reduce parameters and increase efficiency. Specifically, we replace the original dot-product self-attention mechanism with the GATv2 attention mechanism. Additionally, the feed-forward network is removed, which has less negative impact on prediction performance.

**Adaptive Graph Update.** In order to adapt the real cold-start scenarios, inspired by (Hao et al., 2021; 2023), we propose an adaptive graph update pipeline, dynamically updating edges and extracting similar node information according to the current model itself. Formally, we select the top $k_n$ relevant nodes as new neighbors for $d_i$ according to the calculated cosine similarity between drug $d_i$ and $d_j$. Ultimately, we update the biomedical graph $\bar{\mathcal{G}}$ by adding new neighbors to replace the enhanced neighbors for each drug and entity according to similarity, which is then used in the next epoch. The proposed update strategy enables considering the cold-start characteristics of the node neighbors and adaptively selecting appropriate neighbors for message propagation.

**InSMN Module.** Real-world scenarios demonstrate that the significance of representations varies according to different tasks and drugs/entities, which highlights the necessity of the information sharing and interaction to comprehensively exploit drug information. Hence, the InSMN module is proposed to integrate and fuse the fine-level molecular structure knowledge with the semantic and heterogeneous information in the biomedical graph. We first take the molecule-view embedding $\mathbf{H}^{m(l)}$ from the $l$-th GNN layer and the network-view representation $\mathbf{H}^{n(l)}$ of the $l$-th transformer layer as the input. Subsequently, the meta network (Xia et al., 2021; Chen et al., 2023a) is leveraged to extract meta knowledge for preserving important properties from both auxiliary molecule-view and network-view:

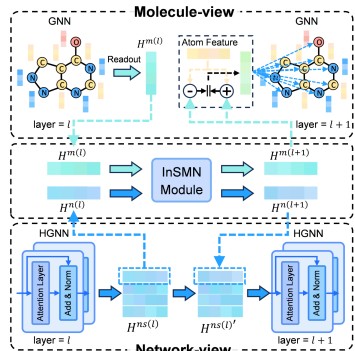

Figure 3: Illustration of the interaction processes between InSMN module and two views.

$$\mathbf{H}^{nm(l)} = \mathbf{H}^{n(l)} \left\| \mathbf{H}^{m(l)} \right\| \sum_{j \in N_{d_i}} \mathbf{h}_{d_i}^{n(l)}, \qquad (2)$$

$$\mathbf{W}_1^{nm(l)} = f_{mlp}^{l_1}\left(\mathbf{H}^{nm(l)}\right), \quad \mathbf{W}_2^{nm(l)} = f_{mlp}^{l_2}\left(\mathbf{H}^{nm(l)}\right), \qquad (3)$$

where $\|$ is concatenation, $\mathbf{H}^{nm(l)} \in \mathbb{R}^{n_d \times 3d}$ denotes the meta knowledge with contextual information for sharing and exchange; $f_{mlp}^{l_1}$ and $f_{mlp}^{l_2}$ are the meta knowledge transformation networks

consisting of two PReLU-activated linear layers. $\mathbf{W}_1^{nm(l)}, \mathbf{W}_2^{nm(l)} \in \mathbb{R}^{n_d \times d \times k_i}$ denote the sharing matrices containing the unique characteristics of drugs, where the transformation rank is restricted as $k_i < d$ to reduce parameters and boost the stability. The ultimate cross-view sharing embeddings for two views are created through fusion as

$$\mathbf{H}^{m(l+1)} = \sigma \left( \mathbf{W}_1^{nm(l)} \mathbf{W}_2^{nm(l)} \mathbf{H}^{m(l)} \right) + \mathbf{H}^{m(l)}, \tag{4}$$

$$\mathbf{H}^{n(l+1)} = \sigma \left( \mathbf{W}_1^{nm(l)} \mathbf{W}_2^{nm(l)} \mathbf{H}^{n(l)} \right) + \mathbf{H}^{n(l)}. \tag{5}$$

where $\sigma$ denotes the PReLU activation function. $\mathbf{H}^{m(l+1)}$ and $\mathbf{H}^{n(l+1)}$ are used as the inputs of the next layer in the GNN encoder and transformer module, respectively. For the molecule-view module, let $\mathbf{h}_{d_i}^{(l)}$ be the node feature vector of drug $d_i$ in the $l$-th GNN layer, we mix the network-view information into its molecular representation learning process as follows (Bi et al., 2023):

$$\mathbf{h}_{d_i}^{(l+1)} = \left[ \mathbf{h}_{d_i}^{(l)} \oplus \mathbf{h}^{m(l+1)} \right] \| \left[ \mathbf{h}_{d_i}^{(l)} \ominus \mathbf{h}^{m(l+1)} \right]. \tag{6}$$

where $\oplus$ and $\ominus$ denote element-wise addition and subtraction operations, respectively. The drug molecules can adaptively acquire coarse-level network link information through this manner for enhancing representation learning ability. With respect to the network-view, $\mathbf{H}^{n(l+1)}$ is directly leveraged to substitute $\mathbf{H}^{ns(l)}[0]$ of the sampled neighborhood sequences at the $l$-th transformer module to produce $\mathbf{H}^{ns(l)'}$, which is then used as input of the $(l+1)$-th transformer module. The interaction processes between InSMN module and two views are illustrated in Figure 3.

## 4.3 CROSS-VIEW GRAPH CONTRASTIVE LEARNING

**Contrastive Loss.** The obtained drug representations from two views are fed into a three-layer linear layer with ReLU activation function to map to the common space, denoted as $\tilde{\mathbf{H}}^m$ and $\tilde{\mathbf{H}}^n$, respectively. The InfoNCE-based contrastive loss is defined for the alignment of two view embeddings:

$$\mathcal{L}_{d_i}^m = -\log \frac{\sum_{d_j \in \mathbb{P}_{d_i}} exp \left( cos \left( \tilde{\mathbf{h}}_{d_i}^m, \tilde{\mathbf{h}}_{d_j}^n \right) / \tau \right)}{\sum_{d_k \in \left\{ \mathbb{P}_{d_i} \cup \mathbb{N}_{d_i} \right\}} exp \left( cos \left( \tilde{\mathbf{h}}_{d_i}^m, \tilde{\mathbf{h}}_{d_k}^n \right) / \tau \right)}, \tag{7}$$

$$\mathcal{L}_{d_i}^n = -\log \frac{\sum_{d_j \in \mathbb{P}_{d_i}} exp \left( cos \left( \tilde{\mathbf{h}}_{d_i}^n, \tilde{\mathbf{h}}_{d_j}^m \right) / \tau \right)}{\sum_{d_k \in \left\{ \mathbb{P}_{d_i} \cup \mathbb{N}_{d_i} \right\}} exp \left( cos \left( \tilde{\mathbf{h}}_{d_i}^n, \tilde{\mathbf{h}}_{d_k}^m \right) / \tau \right)}, \tag{8}$$

$$\mathcal{L}_{con} = \frac{1}{|V_d|} \sum_{d_i \in V_d} \left[ \lambda \cdot \mathcal{L}_{d_i}^m + (1 - \lambda) \cdot \mathcal{L}_{d_i}^n \right]. \tag{9}$$

where $cos(\cdot)$ denotes the cosine similarity function, $\tau$ and $\lambda$ are temperature parameters. $\mathbb{P}_{d_i}$ and $\mathbb{N}_{d_i}$ represent the corresponding positive and negative sample sets for $d_i$, respectively.

**Diffusion Loss.** The above molecule- and network-view representation learning modules capture drug embeddings at different granularities, where the molecule-view learns information from a single molecular structure while the network-view extracts biomedical knowledge based on the heterogeneous topological attribute. To this end, we propose to employ the diffusion model to exchange information between the two different views, bridging the gap between molecule and network representations, and aiding the contrast process. Following the DDPM (Ho et al., 2020) framework, we inject Gaussian noises into the acquired representations $\mathbf{H}^m$ and $\mathbf{H}^n$ for corruption. Concretely, the noises are added into each representation as:

$$q \left( \mathbf{H}_t^m \mid \mathbf{H}_{t-1}^m \right) = \mathcal{N} \left( \mathbf{H}_t^m, \sqrt{1 - \beta_t} \mathbf{H}_{t-1}^m, \beta_t \mathbf{I} \right), \tag{10}$$

where $t \in \{1, 2, \dots, T\}$ denotes the current diffusion step, $\beta_t$ is the variance schedule of the step $t$, $\mathbf{I}$ is the identity matrix, and $\mathcal{N}$ denotes the Gaussian distribution which $\mathbf{H}_t^m$ is sampled from. The reverse process aims to iteratively recover the corrupted embedding $\mathbf{H}_t^m$ through the denoising module, where the neural network model is required to approximate the reverse distribution:

$$p \left( \mathbf{H}_{t-1}^m \mid \mathbf{H}_t^m \right) = \mathcal{N} \left( \mathbf{H}_{t-1}^m; \mu_\theta \left( \mathbf{H}_t^m, t \right), \Sigma_\theta \left( \mathbf{H}_t^m, t \right) \right). \tag{11}$$

where $\theta$ represents the learnable parameters of the neural network, $\mu_\theta\left(\mathbf{H}_t^m, t\right)$ and $\Sigma_\theta\left(\mathbf{H}_t^m, t\right)$ are the mean and covariance of the Gaussian distribution. Regarding the denoising network $f_\theta$, motivated by (Zhu et al., 2023; Yang et al., 2023), we utilize GNN to directly predict the embedding, which is more suitable for the non-Euclidean data. Specifically, we construct the drug similarity graph $\mathcal{G}_{ds}$ by selecting top $k_n$ drugs with the highest similarity of representation $\mathbf{H}^n$. Next, the GATv2 is set as the denoising model to reconstruct $\mathbf{H}^n$ from $\mathbf{H}_t^n$, yielding $\widehat{\mathbf{H}}^m$. The recovered representation of the network-view can be generated in the similar way, denoted as $\widehat{\mathbf{H}}^n$. The diffusion loss is formulated as the expected value of the L2 distance to optimize the two diffusion processes:

$$\mathcal{L}_{diff} = \mathbb{E}_q\left[\left\|\mathbf{H}^n - \widehat{\mathbf{H}}^m\right\|^2\right] + \mathbb{E}_q\left[\left\|\mathbf{H}^m - \widehat{\mathbf{H}}^n\right\|^2\right]. \tag{12}$$

### 4.4 OPTIMIZATION OBJECTIVES

The contrastive loss and diffusion loss are combined to jointly optimize our self-supervised cross-view contrastive learning model:

$$\mathcal{L}_{ssl} = \gamma \cdot \mathcal{L}_{con} + (1 - \gamma) \cdot \mathcal{L}_{diff}. \tag{13}$$

where $\gamma$ is the weighting coefficients. The trained DrugXAS model is then fine-tuned for the downstream link prediction task, where the fine-tuned drug representation $\mathbf{H}^n$ and entity representation $\mathbf{H}^e$ are utilized to produce the link pair embedding via the dot product operation. The final probability of the corresponding link existence is output through a three-layer MLP. We use the binary cross-entropy loss as the optimization objective for link prediction.

## 5 EXPERIMENTS

### 5.1 EXPERIMENTAL SETUP

**Datasets.** To evaluate the performance of DrugXAS, we focus on five types of drug-related biomedical link prediction tasks, including predicting DTIs, drug-disease associations (DDAs), DSEs, miRNA-drug resistance associations (MDAs) and circRNA-drug sensitivity associations (CDAs). Seven widely used benchmark datasets are employed, i.e., LuoDTI (Luo et al., 2017) and ZhengDTI (Zheng et al., 2018) for DTI prediction, LiangDDA (Liang et al., 2017) and ZhangDDA (Zhang et al., 2018) for DDA prediction, PauwelsDSE (Pauwels et al., 2011) for DSE prediction, HuangMDA (Huang et al., 2019) for MDA prediction, and DengCDA (Deng et al., 2022) for CDA prediction. The statistic information of these datasets is presented in Appendix B.

**Evaluation Protocols.** We treat the known links as the positive samples and randomly select an equal number of unlinked drug and entity pairs as negative samples (Fu et al., 2021). Five-fold cross-validation is implemented on each dataset to conduct both warm and cold-start experiments. Two evaluation metrics are employed to measure the performance: the area under the receiver operating characteristic curve (AUC) and the area under the precision-recall curve (AUPR). Please refer to Appendix C for the implementation details and hyperparameter setups.

Table 1: Performance comparison in warm scenarios in terms of AUC and AUPR. The best results are bold, and the second-best results are underlined.

| Datasets | Metric | SkipGNN | SiGraC | CSGNN | MVGCN | HOGCN | CGCN | SubKNet | DMGI | HeCo | HGMAE | HERO | DrugXAS |
|---|---|---|---|---|---|---|---|---|---|---|---|---|---|
| LuoDTI | AUC | 0.8817 | 0.8499 | 0.8693 | 0.9046 | 0.8566 | 0.8655 | 0.9180 | 0.8572 | 0.8628 | 0.8859 | 0.8994 | **0.9358** |
| | AUPR | 0.8871 | 0.8613 | 0.8711 | 0.9022 | 0.8690 | 0.8703 | 0.9272 | 0.8721 | 0.8852 | 0.8977 | 0.9017 | **0.9452** |
| ZhengDTI | AUC | 0.8967 | 0.8444 | 0.9234 | 0.9357 | 0.8567 | 0.9250 | 0.9439 | 0.9242 | 0.9401 | 0.9441 | 0.9426 | **0.9702** |
| | AUPR | 0.8942 | 0.8256 | 0.9176 | 0.9312 | 0.8595 | 0.9183 | 0.9408 | 0.9226 | 0.9470 | 0.9429 | 0.9388 | **0.9695** |
| LiangDDA | AUC | 0.7460 | 0.7211 | 0.7989 | 0.8259 | 0.7148 | 0.8131 | 0.7686 | 0.8346 | 0.8174 | 0.8416 | 0.8622 | **0.9184** |
| | AUPR | 0.7578 | 0.7390 | 0.7994 | 0.8405 | 0.7395 | 0.8165 | 0.7995 | 0.8469 | 0.8445 | 0.8523 | 0.8740 | **0.9216** |
| ZhangDDA | AUC | 0.7493 | 0.8135 | 0.8020 | 0.8439 | 0.7567 | 0.8007 | 0.8420 | 0.7628 | 0.7921 | 0.8091 | 0.8146 | **0.8630** |
| | AUPR | 0.7355 | 0.7716 | 0.8006 | 0.8385 | 0.7462 | 0.7948 | 0.8417 | 0.7652 | 0.7928 | 0.8032 | 0.8127 | **0.8585** |
| PauwelsDSE | AUC | 0.9036 | 0.8889 | 0.9384 | 0.9294 | 0.9125 | 0.9409 | 0.9318 | 0.9118 | 0.9224 | 0.9208 | 0.9328 | **0.9451** |
| | AUPR | 0.9007 | 0.8698 | 0.9372 | 0.9255 | 0.9107 | 0.9389 | 0.9303 | 0.9129 | 0.9244 | 0.9215 | 0.9315 | **0.9425** |
| HuangMDA | AUC | 0.9378 | 0.9184 | **0.9701** | 0.9442 | 0.9366 | 0.9681 | 0.9366 | 0.9175 | 0.8896 | 0.9024 | 0.9257 | 0.9535 |
| | AUPR | 0.9343 | 0.9005 | **0.9671** | 0.9377 | 0.9337 | 0.9649 | 0.9361 | 0.9240 | 0.9012 | 0.9007 | 0.9246 | 0.9477 |
| DengCDA | AUC | 0.7825 | 0.7882 | 0.8711 | 0.8791 | 0.7732 | **0.8892** | 0.8810 | 0.8446 | 0.8659 | 0.8531 | 0.8680 | 0.8812 |
| | AUPR | 0.7831 | 0.7587 | 0.8769 | 0.8820 | 0.7718 | **0.8971** | 0.8828 | 0.8462 | 0.8744 | 0.8548 | 0.8752 | 0.8835 |

**Baselines.** We compare our proposed DrugXAS with two categories of state-of-the-art methods: (1) biomedical link prediction methods, including SkipGNN (Huang et al., 2020), SiGraC (Coşkun & Koyutürk, 2021), CSGNN (Zhao et al., 2021), MVGCN (Fu et al., 2021), HOGCN (KC et al., 2022), CGCN (Shen et al., 2024) and SubKNet (Li et al., 2024a), (2) self-supervised heterogeneous graph representation learning methods DMGI (Park et al., 2020), HeCo (Wang et al., 2021), HG-MAE (Tian et al., 2023) and HERO (Mo et al., 2024).

## 5.2 PERFORMANCE COMPARISON IN WARM SCENARIOS

We first summarize the results of DrugXAS and baseline methods on seven datasets in warm scenario in Table 1. As shown, our DrugXAS yields the best performance in terms of two metrics across five of seven datasets. Besides, the experimental results yield the following observations: (1) Our DrugXAS consistently achieves significant performance improvements in DTI and DDA prediction tasks. We attribute these improvements to the auxiliary molecule-view and InSMN module. The former plays a major role in DTI and DDA processes, while the latter empowers our model to integrate and interact fine-level drug molecular attributes with the semantic relations of the biomedical links. (2) The performances of DrugXAS in MDA and CDA prediction tasks are relatively mediocre, inferior to some other baselines. We assume the phenomena derives from the lower number of drugs in these two datasets, which are insufficient for our multi-view GCL paradigm to explore discriminative drug embeddings. (3) Compared to the self-supervised heterogeneous graph representation learning methods, our model generally yields better performance, further justifying the effectiveness of the molecule-view representation learning and the information sharing component. (4) With respect to GNN-based models that focus on neighbor aggregation (SkipGNN, MVGCN, and HOGCN), DrugXAS consistently surpasses them across all datasets, indicating the superiority and merit of our proposed dynamic neighborhood sampler and adaptive graph updater in warm scenarios.

## 5.3 PERFORMANCE COMPARISON IN COLD-START SCENARIOS

Table 2: Performance comparison in different cold-start scenarios in terms of AUC and AUPR. The best results are bold, and the second-best results are underlined.

| Datasets | Setting | Metric | SkipGNN | SiGraC | CSGNN | MVGCN | HOGCN | CGCN | SubKNet | DMGI | HeCo | HGMAE | HERO | DrugXAS |
|---|---|---|---|---|---|---|---|---|---|---|---|---|---|---|
| LuoDTI | Cold-drug | AUC | 0.8720 | 0.8962 | 0.8496 | 0.8347 | 0.8260 | 0.8837 | 0.8940 | 0.8494 | 0.8612 | 0.8863 | 0.8908 | **0.9051** |
| | | AUPR | 0.8785 | 0.8977 | 0.8295 | 0.8320 | 0.8419 | 0.8805 | 0.8984 | 0.8471 | 0.8557 | 0.8814 | 0.8922 | **0.9096** |
| | Cold-entity | AUC | 0.7216 | 0.7835 | 0.6363 | 0.5782 | 0.5289 | 0.7262 | 0.8001 | 0.7556 | 0.7663 | 0.8185 | 0.8226 | **0.8497** |
| | | AUPR | 0.6782 | 0.7797 | 0.6117 | 0.5691 | 0.5607 | 0.7235 | 0.8040 | 0.7583 | 0.7779 | 0.8216 | 0.8273 | **0.8731** |
| | Cold-pair | AUC | 0.6589 | 0.6322 | 0.6249 | 0.5593 | 0.5394 | 0.5358 | 0.7198 | 0.6604 | 0.6857 | 0.7359 | 0.7752 | **0.8310** |
| | | AUPR | 0.6341 | 0.6378 | 0.6308 | 0.5372 | 0.5488 | 0.5462 | 0.7126 | 0.6739 | 0.6992 | 0.7444 | 0.7861 | **0.8458** |
| ZhengDTI | Cold-drug | AUC | 0.8101 | 0.7762 | 0.7425 | 0.8462 | 0.7545 | 0.8131 | 0.8564 | 0.8137 | 0.8286 | 0.8284 | 0.8547 | **0.8884** |
| | | AUPR | 0.8279 | 0.7862 | 0.7534 | 0.8475 | 0.7474 | 0.8230 | 0.8439 | 0.8088 | 0.8105 | 0.8161 | 0.8468 | **0.8791** |
| | Cold-entity | AUC | 0.6803 | 0.6976 | 0.5866 | 0.7264 | 0.6464 | 0.6709 | 0.7582 | 0.7621 | 0.7580 | 0.7854 | 0.8012 | **0.8106** |
| | | AUPR | 0.6896 | 0.6824 | 0.6027 | 0.7318 | 0.6456 | 0.6938 | 0.7691 | 0.7560 | 0.7427 | 0.7913 | 0.8053 | **0.8264** |
| | Cold-pair | AUC | 0.5336 | 0.5585 | 0.5116 | 0.5987 | 0.5193 | 0.5274 | 0.6571 | 0.5982 | 0.5462 | 0.6625 | 0.6923 | **0.7012** |
| | | AUPR | 0.5269 | 0.5621 | 0.5175 | 0.5893 | 0.5164 | 0.5209 | 0.6508 | 0.5935 | 0.5409 | 0.6578 | 0.6792 | **0.6989** |
| LiangDDA | Cold-drug | AUC | 0.7020 | 0.7253 | 0.6461 | 0.5527 | 0.6803 | 0.7099 | 0.7497 | 0.6178 | 0.6383 | 0.7096 | 0.7466 | **0.7780** |
| | | AUPR | 0.7276 | 0.7398 | 0.6708 | 0.5759 | 0.7109 | 0.7260 | 0.7711 | 0.6338 | 0.6592 | 0.7172 | 0.7602 | **0.7968** |
| | Cold-entity | AUC | 0.5979 | 0.6215 | 0.6544 | 0.5684 | 0.5767 | 0.6280 | 0.6774 | 0.6007 | 0.6129 | 0.6457 | 0.6687 | **0.8078** |
| | | AUPR | 0.5842 | 0.6084 | 0.6362 | 0.5571 | 0.5558 | 0.6085 | 0.6656 | 0.5981 | 0.6088 | 0.6384 | 0.6572 | **0.8016** |
| | Cold-pair | AUC | 0.5408 | 0.5817 | 0.5494 | 0.5174 | 0.5239 | 0.5521 | 0.6257 | 0.5434 | 0.5671 | 0.6289 | 0.6433 | **0.6922** |
| | | AUPR | 0.5287 | 0.5680 | 0.5318 | 0.5096 | 0.5200 | 0.5472 | 0.6282 | 0.5372 | 0.5535 | 0.6307 | 0.6480 | **0.6764** |
| ZhangDDA | Cold-drug | AUC | 0.6542 | 0.7501 | 0.6925 | 0.7451 | 0.6462 | 0.6833 | 0.7577 | 0.7129 | 0.7056 | 0.7606 | 0.7629 | **0.7668** |
| | | AUPR | 0.6324 | 0.7412 | 0.6923 | 0.7162 | 0.6149 | 0.6827 | 0.7388 | 0.7214 | 0.7101 | 0.7529 | 0.7542 | **0.7628** |
| | Cold-entity | AUC | 0.6806 | 0.7521 | 0.6545 | 0.7655 | 0.6531 | 0.7332 | 0.7746 | 0.7243 | 0.7276 | 0.7886 | 0.7630 | **0.7910** |
| | | AUPR | 0.6755 | 0.7586 | 0.6360 | 0.7454 | 0.6540 | 0.7241 | 0.7619 | 0.7186 | 0.7112 | 0.7762 | 0.7596 | **0.7911** |
| | Cold-pair | AUC | 0.5589 | 0.6867 | 0.5603 | 0.6864 | 0.5302 | 0.5770 | 0.6752 | 0.6360 | 0.6257 | 0.6641 | 0.6527 | **0.7066** |
| | | AUPR | 0.5490 | 0.6802 | 0.5420 | 0.6620 | 0.5349 | 0.5654 | 0.6794 | 0.6328 | 0.6190 | 0.6615 | 0.6534 | **0.7047** |
| PauwelsDSE | Cold-drug | AUC | 0.8860 | 0.8024 | 0.7430 | 0.8230 | 0.7915 | 0.8800 | 0.8785 | 0.8796 | 0.8896 | 0.8822 | 0.8912 | **0.8945** |
| | | AUPR | 0.8829 | 0.7984 | 0.7540 | 0.8318 | 0.7791 | 0.8696 | 0.8694 | 0.8754 | 0.8844 | 0.8809 | 0.8858 | **0.8879** |
| | Cold-entity | AUC | 0.6126 | 0.6847 | 0.7627 | 0.7458 | 0.6425 | 0.7301 | 0.8051 | 0.7285 | 0.7358 | 0.8275 | 0.8334 | **0.8698** |
| | | AUPR | 0.5927 | 0.6697 | 0.7601 | 0.7586 | 0.6219 | 0.7125 | 0.8096 | 0.7172 | 0.7234 | 0.8201 | 0.8332 | **0.8580** |
| | Cold-pair | AUC | 0.5377 | 0.5314 | 0.5356 | 0.5854 | 0.5145 | 0.5409 | 0.8213 | 0.5233 | 0.5484 | 0.7857 | 0.8012 | **0.8440** |
| | | AUPR | 0.5321 | 0.5301 | 0.5361 | 0.5791 | 0.5212 | 0.5389 | 0.8178 | 0.5247 | 0.5463 | 0.7775 | 0.8026 | **0.8356** |
| HuangMDA | Cold-drug | AUC | 0.6959 | 0.7792 | 0.6882 | 0.7131 | 0.6472 | 0.7171 | 0.6931 | 0.7256 | 0.6864 | 0.7444 | 0.7691 | **0.7847** |
| | | AUPR | 0.6720 | 0.7775 | 0.6907 | 0.7151 | 0.6370 | 0.7014 | 0.6856 | 0.7292 | 0.7004 | 0.7526 | 0.7660 | **0.7783** |
| | Cold-entity | AUC | 0.9240 | 0.9211 | 0.9187 | 0.9171 | 0.9167 | 0.9255 | 0.9247 | 0.9194 | 0.9165 | 0.9127 | 0.9208 | **0.9298** |
| | | AUPR | 0.9228 | 0.9195 | 0.9137 | 0.9110 | 0.9137 | 0.9193 | 0.9207 | 0.9153 | 0.9132 | 0.9108 | 0.9182 | **0.9244** |
| | Cold-pair | AUC | 0.5955 | 0.6823 | 0.5789 | 0.7379 | 0.5402 | 0.6287 | 0.7982 | 0.6296 | 0.5568 | 0.7387 | 0.7431 | **0.8254** |
| | | AUPR | 0.5931 | 0.6787 | 0.5873 | 0.7350 | 0.5592 | 0.6331 | 0.7881 | 0.6258 | 0.5517 | 0.7286 | 0.7407 | **0.8109** |
| DengCDA | Cold-drug | AUC | 0.7969 | 0.7529 | 0.7845 | 0.7890 | 0.6729 | 0.7691 | 0.7888 | 0.7785 | 0.7899 | 0.7873 | 0.7848 | **0.8082** |
| | | AUPR | 0.7849 | 0.7508 | 0.7680 | 0.7745 | 0.6530 | 0.7551 | 0.7921 | 0.7759 | 0.7801 | 0.7790 | 0.7790 | **0.7951** |
| | Cold-entity | AUC | 0.6430 | 0.6211 | 0.6667 | 0.7328 | 0.5994 | 0.6951 | 0.7403 | 0.7173 | 0.7375 | 0.7334 | 0.7156 | **0.7478** |
| | | AUPR | 0.5983 | 0.6084 | 0.6418 | 0.7292 | 0.5768 | 0.6619 | 0.7308 | 0.7057 | 0.7250 | 0.7227 | 0.7219 | **0.7321** |
| | Cold-pair | AUC | 0.5277 | 0.5434 | 0.5874 | 0.6430 | 0.5181 | 0.5104 | 0.6221 | 0.5517 | 0.5786 | 0.6142 | 0.6384 | **0.6597** |
| | | AUPR | 0.5199 | 0.5468 | 0.5810 | 0.6317 | 0.5271 | 0.5075 | 0.6189 | 0.5406 | 0.5714 | 0.6070 | 0.6295 | **0.6411** |

As illustrated in Table 2, DrugXAS significantly and consistently outperforms all state-of-the-art models across seven benchmark datasets in all cold-start scenarios. The key observations are as follows: (1) The performance benefits of DrugXAS are more pronounced in cold-pair settings, with improvements ranging from 0.89% to 11.12% in terms of AUC. This further confirms that DrugXAS is capable of learning high-quality, robust embeddings for drugs and entities without known links. (2) GNN-based models which devise novel neighbor aggregators offer unsatisfactory performance, implying that their proposed aggregators are not suitable for cold-start scenarios. These results also justify the effectiveness of the dynamic neighborhood sampler and adaptive graph updater in selecting appropriate and informative neighborhoods. (3) The self-supervised heterogeneous graph representation learning baselines maintain relatively competitive and stable performance, indicating the capacity of self-supervised learning for alleviating the data scarcity of cold-start issue.

## 5.4 ABLATION STUDY

To gain deeper insights into each essential module of DrugXAS, we conduct the ablation study by constructing the following variant models: (A) *w/o-MA:* The molecule-view augmentation scheme is removed in this variant; (B) *w/o-NA:* The network-view augmentation scheme is removed in this variant; (C) *w/o-DNS:* The dynamic neighborhood sam-

Table 3: Performance comparison of the ablation experiment in warm scenarios.

| Datasets | Metric | w/o-MA | w/o-NA | w/o-DNS | w/o-AGU | w/o-InSMN | w/o-Diff | DrugXAS |
|---|---|---|---|---|---|---|---|---|
| LuoDTI | AUC | 0.9142 | 0.9307 | 0.9233 | _0.9324_ | 0.9126 | 0.9320 | **0.9358** |
| | AUPR | 0.9223 | 0.9384 | 0.9363 | _0.9419_ | 0.9168 | _0.9421_ | **0.9452** |
| ZhengDTI | AUC | 0.9482 | 0.9567 | 0.9684 | _0.9692_ | 0.9502 | 0.9651 | **0.9702** |
| | AUPR | 0.9475 | 0.9532 | 0.9652 | _0.9670_ | 0.9473 | 0.9617 | **0.9695** |
| LiangDDA | AUC | 0.9017 | _0.9167_ | 0.9102 | 0.9119 | 0.9028 | 0.9118 | **0.9184** |
| | AUPR | 0.9095 | _0.9207_ | 0.9187 | 0.9168 | 0.9050 | 0.9189 | **0.9216** |
| ZhangDDA | AUC | 0.8609 | 0.8511 | 0.8567 | _0.8621_ | 0.8436 | 0.8563 | **0.8630** |
| | AUPR | _0.8581_ | 0.8474 | 0.8536 | _0.8570_ | 0.8397 | 0.8540 | **0.8585** |
| PauwelsDSE | AUC | 0.9307 | 0.9338 | _0.9409_ | 0.9398 | 0.9220 | 0.9349 | **0.9451** |
| | AUPR | 0.9302 | 0.9319 | _0.9389_ | 0.9372 | 0.9197 | 0.9318 | **0.9425** |
| HuangMDA | AUC | 0.9462 | 0.9467 | _0.9471_ | 0.9447 | 0.9379 | 0.9441 | **0.9535** |
| | AUPR | 0.9408 | 0.9395 | _0.9422_ | 0.9365 | 0.9344 | 0.9357 | **0.9477** |
| DengCDA | AUC | 0.8796 | 0.8787 | _0.8803_ | 0.8785 | 0.8751 | 0.8801 | **0.8812** |
| | AUPR | _0.8831_ | 0.8811 | 0.8810 | 0.8786 | 0.8759 | 0.8812 | **0.8835** |

pler is dropped, sampling neighborhood sequences only once at the beginning of the training; (D) *w/o-AGU:* We do not include the adaptive graph updater, where the biomedical graph is fixed during training and inference; (E) *w/o-InSMN:* The InSMN component is disabled; and (F) *w/o-Diff:* The diffusion loss is excluded, adopting only the contrastive loss in the contrast process.

The comparison results of DrugXAS and variants in warm scenarios are presented in Table 3. From the results, we have the following observations: (1) All modules have different levels of positive contributions to the model performance. For the augmentation strategy, DrugXAS is consistently superior to w/o-MA and w/o-NA, verifying the positive impact of attention-aware augmentation in both two views. (2) The removal of the dynamic neighborhood sampler and adaptive graph updater results in a slight performance reduction. We speculate that it is because the normal neighbor sampling strategy of GNN is sufficient to handle the warm scenario, where all nodes have a certain number of neighbor nodes to message passing. (3) The performance of w/o-InSMN is considerably inferior, reflecting the necessity of information interaction and sharing between different level of drug data. More detailed ablation experiments are provided in Appendix D.

## 5.5 EMBEDDING VISUALIZATION

To verify the effectiveness of the InSMN module, we further compare the learned drug embeddings intuitively. More specifically, we employ t-SNE (Van der Maaten & Hinton, 2008) to visualize the learned drug representations of DrugXAS and w/o-InSMN, classifying drugs according to Anatomical Therapeutic Chemical (ATC) codes. As presented in Figure 4, it can be observed that DrugXAS outperforms its variant w/o-InSMN in grouping 14 types of drugs, with DrugXAS depicting better drug domain alignment capacity and w/o-InSMN displaying blurry boundaries to some degree. For instance, for the 110 cardiovascular and 104 antiin-

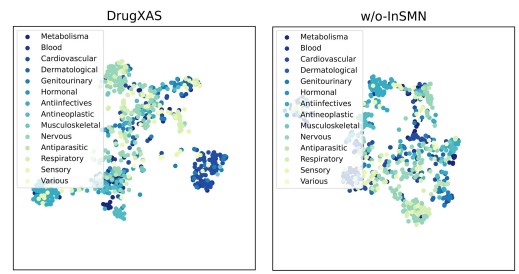

Figure 4: The t-SNE visualization of the learned drug representations on the LiangDDA dataset.

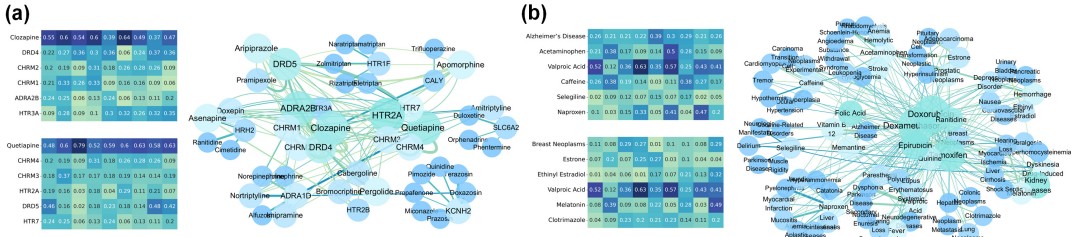

Figure 5: The explanatory subgraphs of case studies. Left: The attention scores of nodes and their sampled neighborhoods. Right: Network visualization of drugs/diseases and the predicted candidate targets/drugs. (a) The interpretability illustration on LuoDTI dataset. (b) The interpretability illustration on ZhangDDA dataset.

fectives drugs, which constitute large proportions of the dataset, DrugXAS exhibits more accurate and separable clusters compared to w/o-InSMN. The results indicate that the discriminative and expressive drug representations encoded by DrugXAS can be primarily attributed to the fine-grained information sharing and interaction provided by InSMN.

## 5.6 INTERPRETABILITY

To explore the in-depth interpretability of DrugXAS, we provide a comprehensive analysis of the novel biomedical link prediction scenario. For the two case study examples in Appendix G, we visualize the target drugs/diseases and predicted proteins/drugs, along with their sampled neighbor nodes within neighborhood sequence and the attention scores, as depicted in Figure 5. We can observe that most predicted candidate proteins/diseases have known interactions/associations with the target drug/disease in their corresponding neighborhood sequence. For example, for the predicted protein DRD5 related to Clozapine, our model infers this novel interaction based on existing DTIs, such as Pramipexole and DRD4, Aripiprazole and CHRM1, which are all sampled neighbor nodes for DRD5 and Clozapine, respectively. Another example is the target HRH2, where its neighbors Asenapine and Doxepin both interact with Clozapine's neighbor targets, DRD4 and ADRA2B, which are also predicted with high attention scores. With respect to the novel link prediction on ZhangDDA dataset, DrugXAS also captures confirmed DDAs to facilitate prediction. For instance, according to the existing DDAs (Folic Acid and Anemia Hemolytic, Tamoxifen and Cardiovascular diseases), Acetaminophen and Ethinyl Estradiol are predicted as candidate drugs for Alzheimer's disease and breast neoplasms, respectively, which align with the known data in the dataset. Through the application of attention-aware augmentation, useful and informative knowledge can be extracted from the neighborhood subgraph. Moreover, the learned attention matrices can be utilized to augment and prune the neighborhood subgraph, thus creating pathways for mechanisms that summarize the biomedical links. To sum up, our investigation verifies and evaluates the generated explanations, underscoring the promising capability of DrugXAS to provide interpretable insights for identifying latent links from a biomedical perspective.

## 6 CONCLUSION

In this paper, motivated to address the limitations of previous work, we propose DrugXAS, an interpretable and adaptive graph contrastive learning framework with information sharing for drug-related biomedical link prediction. DrugXAS introduces the molecule-view learning as an auxiliary aspect to comprehensively exploit biomedical information. The adaptive and dynamic update technique ensures the resilience of DrugXAS in cold-start scenarios, and the augmentation scheme enables achieving interpretability from a biological perspective. A novel information sharing module is devised to jointly capture semantic relations and topological patterns while preserving the chemical properties of drug molecules. In addition, the diffusion model is employed to fill the gap between the two distinct views, enhancing the GCL procedure. Extensive experiments compared with the state-of-the-arts demonstrate the superior effectiveness and robustness of DrugXAS, with competitive interpretability.

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

# APPENDIX

## A   AUGMENTATION STRATEGY

### A.1   MOLECULE-VIEW AUGMENTATION

**Atom masking.**   Atoms/nodes with the smallest attention scores based on a certain ratio are masked, where their node feature vectors are set to 0. Our drug encoder can learn intrinsic chemical properties through the masking operation.

**Bond deletion.**   Bonds/edges with the smallest attention scores based on a certain ratio are deleted, where the edges are removed completely to block the message propagation of GNN encoder. Bond deletion enables our model to discern correlations between a molecule's participation in diverse biomedical reactions.

### A.2   NETWORK-VIEW AUGMENTATION

**Node masking.**   Drugs and entities with the smallest attention scores within neighborhood sequences based on a certain ratio are masked, where their node feature vectors are set to 0. This type of node-level augmentation ensures that the selected node provides the strongest information.

**Edge adding.**   The 2-hop neighbors with the highest attention scores within neighborhood sequences based on a certain ratio are connected to the target node as new neighbors, where the new edges can be added to the subsequent message propagation of HGNN encoder. The idea of this type of augmentation lies in that the original graph structure is not necessarily reliable and the unconnected 2-hop neighbors may contain more structural information.

**Edge deletion.**   The 1-hop neighbors with the smallest attention scores within neighborhood sequences based on a certain ratio are deleted, where the edges are removed completely to block the message propagation of HGNN encoder. Our goal of this component is to judge and reduce noise, such as incomplete or insufficient biomedical interaction data, to enable meaningful and effective integration of information.

## B   DATASETS

Table 4: Statistics of seven experimented datasets

| Dataset | Type | #Drugs | #Entities | #Edges | Sparsity |
|---|---|---|---|---|---|
| LuoDTI | DTI | 708 | 1,512 | 1,923 | 0.9982 |
| ZhengDTI | DTI | 1,094 | 1,556 | 11,819 | 0.9931 |
| LiangDDA | DDA | 763 | 681 | 3,051 | 0.9941 |
| ZhangDDA | DDA | 269 | 598 | 18,416 | 0.8855 |
| PauwelsDSE | DSE | 888 | 1,385 | 61,102 | 0.9503 |
| HuangMDA | MDA | 106 | 754 | 3,338 | 0.9582 |
| DengCDA | CDA | 218 | 271 | 4,134 | 0.9300 |

## C   IMPLEMENTATION DETAILS AND HYPERPARAMETERS

The proposed DrugXAS is optimized by Adam with the learning rate searching from 1e-4 to 2e-3. We apply the ReduceLROnPlateau scheduler of Pytorch and the early stopping strategy to prevent overfitting. The embedding dimension $d$ is set to 128. We set the molecule-view GNN encoder layer to 2, the network-view local structure encoder layer to 2, and the HGNN encoder layer to 2. The coefficients $\lambda$ and $\gamma$ are set to 0.5 and 1, respectively. The search space of the low-rank $k_i$ in InSMN is $\{1, 2, 3, 5, 8, 10\}$. Minimum and maximum augmentation ratios are tuned from 0.1 to

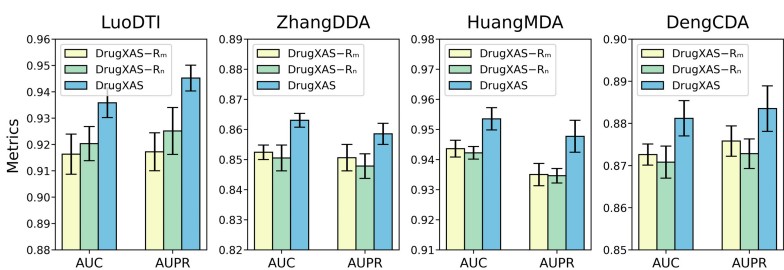

Figure 6: Comparison results of different graph augmentation strategy.

0.5. The number of similarity neighbors $k_n$ and the length of neighborhood sequence $s_n$ are ranged from $\{1, 2, 3, 5\}$ and $\{3, 4, 5, 6, 8, 10\}$, respectively. Table 5 elaborates the detailed hyperparameter settings for each dataset of DrugXAS.

Table 5: Specified hyperparameters of each dataset.

| Dataset | epochs | lr | $d$ | $p_0$ | $p_x$ | $k_n$ | $s_n$ | $k_i$ | $\tau$ | $\lambda$ | $\gamma$ |
|---|---|---|---|---|---|---|---|---|---|---|---|
| LuoDTI | 1,000 | 2e-4 | 128 | 0.3 | 0.4 | 3 | 6 | 5 | 0.8 | 0.5 | 0.5 |
| ZhengDTI | 1,000 | 2e-4 | 128 | 0.2 | 0.3 | 3 | 5 | 5 | 0.8 | 0.5 | 0.5 |
| LiangDDA | 1,000 | 2e-4 | 128 | 0.1 | 0.3 | 3 | 6 | 5 | 0.8 | 0.5 | 0.5 |
| ZhangDDA | 1,500 | 5e-4 | 128 | 0.2 | 0.3 | 5 | 10 | 8 | 0.8 | 0.5 | 0.5 |
| PauwelsDSE | 2,000 | 5e-4 | 128 | 0.2 | 0.5 | 5 | 10 | 10 | 0.8 | 0.5 | 0.5 |
| HuangMDA | 1,000 | 2e-4 | 128 | 0.1 | 0.3 | 3 | 5 | 5 | 0.8 | 0.5 | 0.5 |
| DengCDA | 1,000 | 2e-4 | 128 | 0.2 | 0.4 | 3 | 6 | 5 | 0.8 | 0.5 | 0.5 |

# D  ADDITIONAL ABLATION STUDY

## D.1  EFFECT OF ATTENTION-AWARE AUGMENTATION STRATEGY

To explore the impact of our attention-aware augmentation pipelines, we compare the performance of DrugXAS with the random augmentation strategy, which randomly augments molecular graphs and the biomedical graph using the same augmentation method and ratio: (A) ***DrugXAS-$R_m$:*** Atoms and Bonds in the molecule-view are augmented randomly based on a certain ratio, ignoring the learned attention scores; and (B) ***DrugXAS-$R_n$:*** Nodes and edges in the network-view are augmented randomly based on a certain ratio, ignoring the learned attention scores.

As illustrated in Figure 6, we draw the following conclusions: (1) Both random augmentation strategies of the two views display varying degrees of performance degradation, verifying the contribution of our attention-aware graph augmentation methods. The random perturbation approach might disrupt the molecular structure and introduce meaningless noise for the biomedical links, resulting in their negative impacts. Our devised graph augmentation methods alleviate this problem by using attention scores as an indicator for providing the model with more effective information. (2) DrugXAS-$R_m$ yields outperformance over DrugXAS-$R_n$ in the majority of instances, emphasizing that semantic knowledge is more essential for these types of link prediction tasks. Nevertheless, DrugXAS-$R_m$ exhibits inferior performance for DTI prediction, which is consistent with our assumption that molecular structure information is more beneficial for this type of task.

## D.2  EFFECT OF DYNAMIC NEIGHBORHOOD SAMPLING AND ADAPTIVE GRAPH UPDATE

Our motivation to construct the dynamic neighborhood sampling and adaptive graph update component is to enhance the representation learning capability for cold-start scenarios. Here we investigate whether our proposed two modules can improve performance under cold-start settings to alleviate the issue. To this end, we compare the performance of DrugXAS and two variants, w/o-DNS and w/o-AGU, in cold-start scenarios. According to the results reported in Figure 7, the following find-

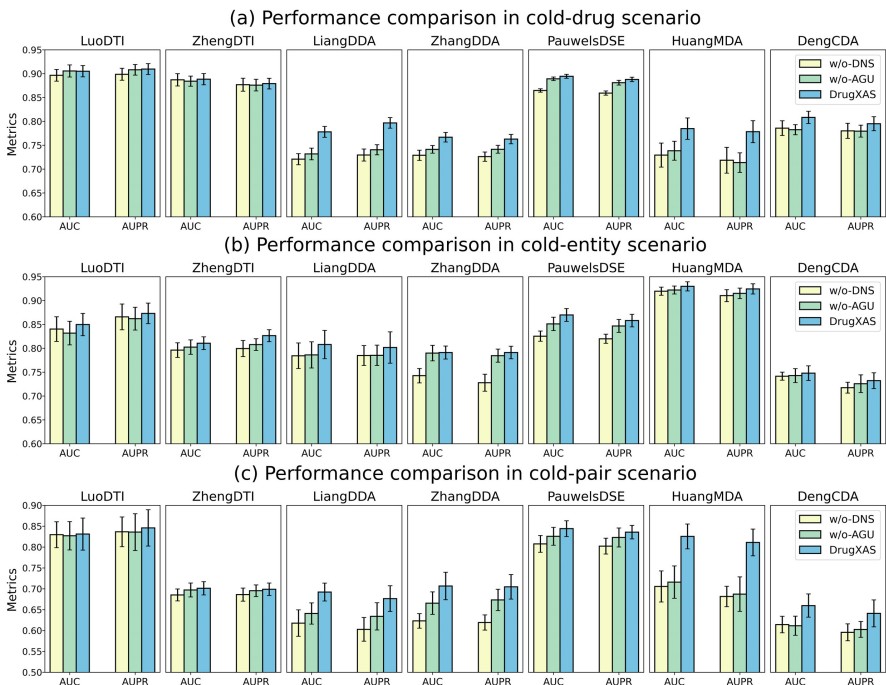

Figure 7: Comparison results of dynamic neighborhood sampling and adaptive graph update in cold-start scenarios.

ings can be made: (1) DrugXAS outperforms both w/o-DNS and w/o-AGU in all cold-start cases, reflecting the rationality of introducing dynamism and adaptivity into cold-start scenarios, which can facilitate our model to adaptively identify neighbors based on the encoder performance and thus enhancing the aggregation ability. (2) The two variants exhibit similar performances, with w/o-AGU slightly exceeding w/o-DNS in the majority of cases, indicating that the dynamic neighborhood sampling is more imperative and effective. This module empowers our DrugXAS to aggregate node information from different high-order neighbors, thereby improving the generalization capability by generating robust embeddings for unseen nodes.

Furthermore, we provide an example on LuoDTI dataset to understand and illustrate how the dynamic neighborhood sampler and adaptive graph updater sample appropriate neighbors for cold-start nodes. As depicted in Figure 8, under the cold-drug setting, for the cold drug node Ibutilide, our proposed dynamic neighborhood sampler samples six drugs. Among them, Pimozide, Terazosin, Amlodipine and Nicardipine are relevant high-order neighbor drugs that interact with target proteins of Ibutilide, also demonstrating a high degree of similarity. Moreover, Amlodipine and Nicardipine are both cardiovascular drugs with the identical ATC codes as Ibutilide, causing them to be updated by the adaptive graph updater to Ibutilide's new neighbors. The results indicate that our devised neighborhood sampler and graph updater adaptively selects the high-order neighbors with informative and effective knowledge. However, the GNN-based baselines ignore the cold-start characteristics of the neighbors in the graph convolution process, failing to specifically handle cold-start neighbor nodes.

# E  HYPERPARAMETER ANALYSIS

In this part, we investigate the performance variation of DrugXAS on several key hyperparameters.

## E.1  IMPACT OF AUGMENTATION RATIO $p_0$ AND $p_x$

We perform the grid search experiments on the initial augmentation ratio $p_0$ and maximum augmentation ratio $p_x$, which are both selected from $\{0.1, 0.2, 0.3, 0.4, 0.5\}$, respectively. Due to the

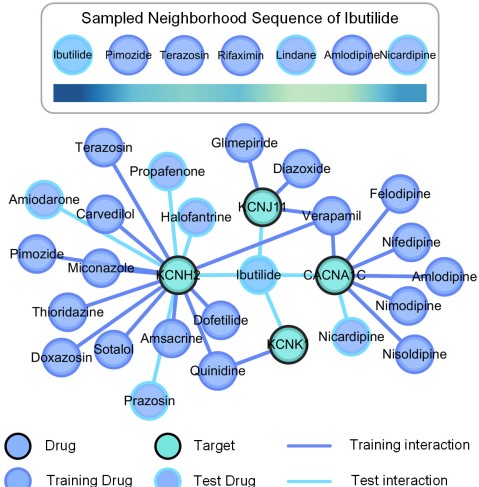

Figure 8: An example of the dynamic neighborhood sampling and adaptive graph update in the cold-start scenario.

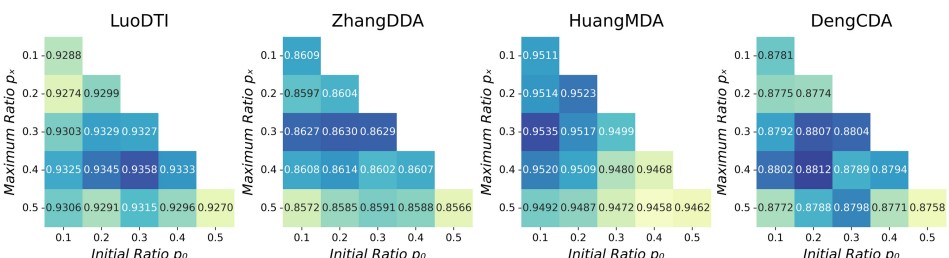

Figure 9: Hyperparameter analysis of the augmentation ratio.

page limit, we only report the AUC metric results since the AUPR metric shows the same trend. As shown in Figure 9, with the increase of $p_0$ and $p_x$, the performance displays a similar trend of rising and then dropping. DrugXAS achieves the best performance on all datasets when both ratios are in the range of $\{0.2, 0.3, 0.4\}$. These results imply that a low augmentation ratio is not capable of generating effective augmented graphs, while an excessive ratio leads to the destruction of the original chemical and semantic information, corrupting the learned node representations.

### E.2 IMPACT OF $k_n$ AND $s_n$

This experiment is conducted to investigate the impact of our dynamic and adaptive HGNN model by varying the combinations of similarity neighbor number $k_n$ and sequence length $s_n$, with the set $\{1, 2, 3, 5\}$ for $k_n$ and $\{3, 4, 5, 6, 8, 10\}$ for $s_n$. The results are reported in Figure 10, from which we note that both too small and too large $k_n$ and $s_n$ cause performance degradation. One possible explanation is that when these two hyperparameters are too low, the sampled similar node and neighbor are not informative enough to generate discriminative and expressive embeddings. By contrast, high values of these parameters introduce noisy high-order neighbors with less meaningful information, confusing and weakening message propagation capability of the HGNN encoder. Additionally, it is observed that the optimal values of $k_n$ and $s_n$ vary across different datasets, which can be attributed to the varying sparsity and distribution, where the distribution of node degrees and edges affects the performance.

### E.3 IMPACT OF LOW-RANK TRANSFORMATION DIMENSION $k_i$

Here, we test the sensitivity of the low-rank dimension $k_i$ of InSMN from 1 to 10. As presented in Figure 11, we observe that increasing $k_i$ doesn't guarantee an improved outcome for all datasets,

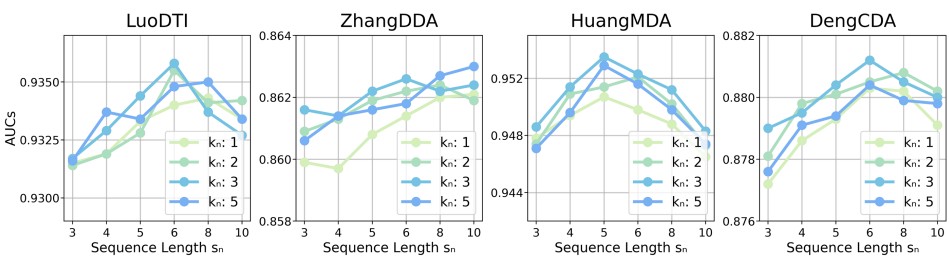

Figure 10: Hyperparameter analysis of $k_n$ and $s_n$.

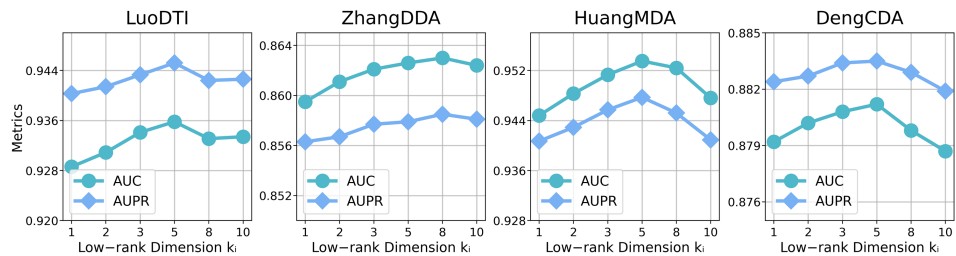

Figure 11: Hyperparameter analysis of the low-rank transformation dimension $k_i$.

and the optimal performance is reached when $k_i = 5$ or $8$. The performance becomes stable when $k_i$ is in the range of $5$ to $8$, and a relatively large value of $k_i$ is required for the high-density ZhangDDA dataset. In general, our model performance reaches its optimum with an appropriate low-rank dimension, while excessive values result in increased training resources and time, decreasing link prediction.

## F EMBEDDING VISUALIZATION

To provide an intuitive understanding and evaluation, we visualize the t-SNE embeddings of positive and negative drug-entity pairs under the warm scenario setting. The node pair embeddings learned by CSGNN, MVGCN, CGCN, SubKNet, and DrugXAS on the ZhangDDA dataset are projected into a 2-dimensional space, as depicted in Figure 12. According to the figure, we can observe that CSGNN, MVGCN, CGCN, and SubKNet have the tendency to separate the positive (blue points) and negative samples (green points), while their embeddings are still mixed to a certain extent with blurry boundaries. In contrast, DrugXAS exhibits the best embedding alignment apparently, correctly separating positive and negative pairs with relatively clear and distinct boundaries. Furthermore, we provide a quantitative analysis by calculating the silhouette scores of the clusters, and our DrugXAS also outperforms other baseline methods with the highest silhouette score.

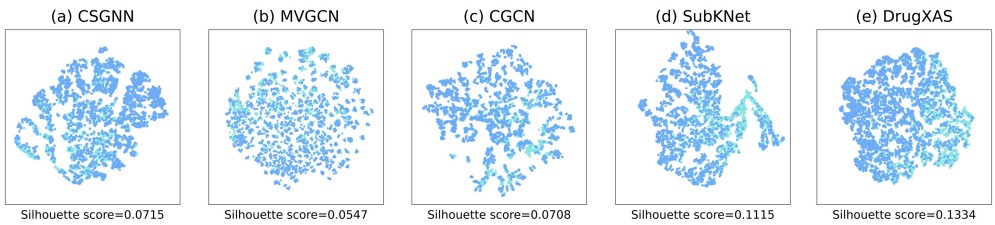

Figure 12: The t-SNE visualization of the learned pair embeddings on the ZhangDDA dataset.

Table 6: The top 5 candidate targets predicted by DrugXAS for two drugs on LuoDTI dataset.

| Drug | Rank | Target ID | Target Name | Evidence |
|---|---|---|---|---|
| Clozapine | 1 | P21918 | DRD5 | DrugBank |
| | 2 | P30939 | HTR1F | DrugCentral |
| | 3 | P25021 | HRH2 | DrugCentral |
| | 4 | P41595 | HTR2B | DrugCentral |
| | 5 | P25100 | ADRA1D | DrugCentral |
| Quetiapine | 1 | P30939 | HTR1F | DrugCentral |
| | 2 | Q9NYX4 | CALY | DrugBank |
| | 3 | P23975 | SLC6A2 | DrugCentral |
| | 4 | Q12809 | KCNH2 | DrugCentral |
| | 5 | P41595 | HTR2B | DrugCentral |

Table 7: The top 5 candidate drugs predicted by DrugXAS for two diseases on ZhangDDA dataset.

| Disease | Rank | Drug ID | Drug Name | Evidence (PMID) |
|---|---|---|---|---|
| Alzheimer's Disease | 1 | DB00316 | Acetaminophen | 19291322 |
| | 2 | DB00313 | Valproic Acid | 19748552 |
| | 3 | DB00201 | Caffeine | 24780254 |
| | 4 | DB01037 | Selegiline | 8998375 |
| | 5 | DB00788 | Naproxen | 21504739 |
| Breast Neoplasms | 1 | DB00655 | Estrone | 29660508 |
| | 2 | DB00977 | Ethinyl Estradiol | 7226163 |
| | 3 | DB00313 | Valproic Acid | 30075223 |
| | 4 | DB01065 | Melatonin | 26292662 |
| | 5 | DB00257 | Clotrimazole | 22347377 |

## G  CASE STUDY

The case study section aims to validate the practical capability of DrugXAS in discovering novel biomedical links. We deploy DrugXAS on LuoDTI and ZhangDDA datasets, constructing training data with all known DTIs/DDAs and establishing test samples with all unknowing pairs. For LuoDTI dataset, two drugs are selected for the case study, i.e., Clozapine and Quetiapine, which have the largest number of existing interactions in the dataset. The top 5 predicted candidate targets with the highest prediction probabilities are presented in Table 6. We can observe that all predicted potential targets can be confirmed in DrugBank or DrugCentral databases, with 2 candidate proteins reported by DrugBank and 8 candidate proteins identified by DrugCentral, implying the veracity of interactions between these predicted targets and two chosen drugs. Regarding ZhangDDA dataset, Alzheimer's disease and breast neoplasms are chosen to conduct the case study due to the thorough profile of their known drugs in the dataset. The top 5 candidate drugs predicted by DrugXAS for the two diseases are listed in Table 7, showcasing that all predicted drugs are supported by corresponding literature. In summary, the case studies further demonstrate the potential of DrugXAS as a promising tool for discovering novel biomedical links.

