# OpenReview forum: "Interpretable and Adaptive Graph Contrastive Learning with Information Sharing for Biomedical Link Prediction"
_ICLR.cc/2025/Conference — ICLR 2025 Conference Withdrawn Submission_

### Official Review · Reviewer_ePJ2 · 2024-11-01

**Soundness:** 3
**Presentation:** 3
**Contribution:** 2
**Rating:** 5
**Confidence:** 4

**Summary:**

This paper proposes DrugXAS, an interpretable and adaptive graph contrastive learning framework with information sharing for drug-related biomedical link prediction. It addresses limitations in existing solutions by introducing three key features: interpretability, robustness, and precision.

**Strengths:**

The topic is interesting and valuable. The experimental results look positive. The paper is technically sound and well-structured.

**Weaknesses:**

The novelty of this paper is limited. The core idea of the DrugXAS solution is similar to SNRI [1] and BioKDN [2] (e.g., the structure learning of local subgraphs). The authors can provide more discussion to clarify the key differences between them.

In the ablation study, why did the dynamic neighborhood sampler not show significant improvements? Is the dynamic neighborhood sampler necessary? The authors can conduct more experiments to explore the reasons behind this.

In section 5.5, the visualizations of drug embeddings for DrugXAS and DrugXAS w/o-SMN show no significant difference for most drug types. The authors can discuss further reasons.

[1] Xu, X., Zhang, P., He, Y., Chao, C., & Yan, C. (2022). Subgraph neighboring relations infomax for inductive link prediction on knowledge graphs. AAAI 2022.

[2] Ma, T., Chen, Y., Tao, W., Zheng, D., Lin, X., Pang, C. I., ... & Philip, S. Y. (2024). Learning to Denoise Biomedical Knowledge Graph for Robust Molecular Interaction Prediction. IEEE Transactions on Knowledge and Data Engineering.

**Questions:**

see the weaknesses.

---

### Official Review · Reviewer_zmag · 2024-11-01

**Soundness:** 3
**Presentation:** 2
**Contribution:** 1
**Rating:** 3
**Confidence:** 4

**Summary:**

The paper presents DrugXAS, a novel framework aimed at predicting links in biomedical networks, particularly drug-related links.
DrugXAS addresses three main challenges in the field: interpretability, robustness and flexibility in cold-start scenarios, and integration of multi-view information.
The framework incorporates an attention-aware augmentation strategy for interpretability, an adaptive graph updater and neighborhood sampler to enhance representation learning in cold-start situations, and an information-sharing module with diffusion loss to fuse molecular and network data for contrastive learning.
Experimental results on seven datasets show that DrugXAS outperforms state-of-the-art methods across various metrics, demonstrating improved prediction accuracy, interpretability, and adaptability.

**Strengths:**

- Introduces a novel end-to-end framework that creatively combines attention-based augmentation, adaptive graph updates, and information sharing between molecular and network views
- The attention-aware augmentation strategy is a new approach to providing interpretability in this domain
- The information sharing module presents an original way to bridge molecular-level and network-level drug information
- Comprehensive experimental evaluation on 7 different benchmark datasets covering multiple types of biomedical link prediction tasks

**Weaknesses:**

1. Computational Efficiency and Scalability (Major)

2. Constrained Evaluation of Interpretability

3. Augmentation Strategy Issues

4. Comparison with Alternative Contrastive Learning Approaches

**Questions:**

**1. Computational Efficiency and Scalability**
The paper lacks crucial analysis of computational overhead and scalability.
While the model achieves good performance in smaller datasets, it introduces multiple complex components (attention-based augmentation, dynamic neighborhood sampling, information sharing module) without analyzing their computational costs.
- No runtime comparisons are provided against simpler baselines.
- The memory requirements for storing attention matrices and neighborhood sequences could be substantial for larger graphs.

So, the paper should include complexity analysis, memory profiling, and scaling experiments on larger datasets to demonstrate real-world applicability.
This is my major concern for the paper.

**2. Constrained Evaluation of Interpretability**
- The interpretability claim of DrugXAS is primarily based on qualitative results, relying heavily on case studies without systematic evaluations.
- Most importantly, there is no validation from domain experts to confirm whether the generated explanations are meaningful from a biomedical perspective.
A structured human evaluation involving pharmacologists or medical researchers would significantly strengthen these claims.
- Furthermore, it remains contentious whether attention provides a valid explanation [1.2]

**3. Augmentation Strategy Issues**
The attention-aware augmentation strategy introduces additional hyperparameters (p0, px) whose selection seems crucial but poorly justified.
Additionally, limited analysis of how the augmentation impacts different tasks and dataset types limits our understanding of its broader applicability and potential areas for improvement.

**4. Comparison with Alternative Contrastive Learning Approaches**
Similar to the Augmentation Strategy Issues, the paper's contrastive learning component lacks comprehensive comparison with recent advances in the field.
While some baseline methods are included, notable omissions include momentum contrast approaches, multi-view contrastive methods, and hierarchical contrastive learning frameworks.
The paper should also explore how different positive/negative sampling strategies affect performance, particularly in the cold-start scenario where negative sampling becomes more challenging.


**Minor**
- What do these grey edges mean in Figure 1 (b)?
-  Many characters in Figure 2 are hard to recognise.

[1] Attention is not explanation

[2] Is attention explanation? an introduction to the debate

---

### Official Review · Reviewer_o3Bw · 2024-11-01

**Soundness:** 3
**Presentation:** 3
**Contribution:** 2
**Rating:** 3
**Confidence:** 4

**Summary:**

This paper proposes an interpretable and adaptive contrastive learning model, DrugXAS, which effectively address three main challenges in drug-related biomedical link prediction. Leveraging attention-aware augmentation scheme, an adaptive graph updater and neighborhood sampler, and an information sharing module with diffusion loss, this paper learns molecule- and network-view representations, effectively integrating topological and chemical information in biomedical networks. Extensive experiments show that the proposed DrugXAS outperforms the state-of-the-art baselines on seven benchmark datasets for different types of tasks.

**Strengths:**

1. The paper introduces attention-aware augmentation scheme, an adaptive graph updater and neighborhood sampler, and an information sharing module with diffusion loss, effectively integrating chemical and topological information from both molecule- and network-view representations. This improves the effectiveness and interpretability in biomedical link prediction.
2. The paper conducts extensive experiments on seven benchmark datasets for different types of tasks, demonstrating the proposed DrugXAS's superior performance over state-of-the-art baselines under both warm and cold-start scenarios.
3. The paper is well-written and organized, with a clear problem formulation, methodology, and experimental evaluation.

**Weaknesses:**

1. Limited novelty. This paper introduces a multi-view augmentation scheme, GNN encoder, and graph contrastive learning, which have been widely used in many relevant works [1,2,3,4]. Although DrugXAS proposes an adaptive graph updater and neighborhood sampler to enhance aggregation ability, the work does not seem to provide more substantial technical contributions or novel insights.
2. Insufficient experiment evaluation. The proposed DrugXAS includes several hyperparameters, such as the weighting coefficients $\gamma$ in Eq. 13 and the node representations dimension $d$, which require further assess their impact on biomedical link prediction. Moreover, it is important to change different GNN encoders for molecule- and network-view representation learning to verify the model's effectiveness and robustness.
3. Lack of theoretical analysis. This paper constructs the enhanced graph by selecting the top $k_n$ nodes with the highest cosines similarity for each node, and adaptively updates the graph for better adaptation to cold-start scenarios. However, it is necessary to demonstrate this effectiveness of this strategy, including the selection of $k_n$, via a thorough theoretical analysis.

**Reference**
[1] MDDI-SCL: predicting multi-type drug-drug interactions via supervised contrastive learning, Journal of Cheminformatics, 2020.
[2] Molecular Contrastive Learning of Representations via Graph Neural Networks, nature machine intelligence, 2022.
[3] DSN-DDI: an accurate and generalized framework for drug-drug interaction prediction by dual-view representation learning, Brief Bioinform, 2023.
[4] DIG-Mol: A Contrastive Dual-Interaction Graph Neural Network for Molecular Property Prediction, Journal of Biomedical and Health Informatics, 2024.

**Questions:**

1. Section 3 defines the biomedical heterogeneous graph as an undirected graph, yet Figure 1 includes light blue arrows. What do these arrows represent?
2. Since the proposed DrugXAS has many hyperparameters (e.g., $\gamma$, $k_n$, $s_n$,and $p_0$), it is inefficient and impractical to select different hyperparameters tailored to different datasets. Could the authors clarify how they address this issue?
3. According to the proposed dynamic neighborhood sampling in Section 4.2, the DrugXAS iteratively add its neighbors from 1-hop to higher hop to the neighborhood sequence until the sequence length reaches $s_n$. If the number of neighbors from 1-hop exceeds $s_n$, how do the authors select which neighbors to sample?
4. Could the authors further provide a comprehensive analysis of the model interpretability on various downstream tasks under both warm and cold-start scenarios, which is valuable for understanding the precision and utility of DrugXAS in healthcare applications?

---

### Official Review · Reviewer_Kag3 · 2024-11-03

**Soundness:** 3
**Presentation:** 2
**Contribution:** 2
**Rating:** 3
**Confidence:** 3

**Summary:**

The paper introduces DrugXAS, a framework for predicting unobserved links in drug-related biomedical networks, enhancing drug discovery and molecular understanding. DrugXAS addresses several challenges: (1) interpretability, (2) robustness for cold start, and (3) capturing multi-view information, through incorporating (1) attention-aware augmentation, (2) adaptive graph updating, and (3) an information-sharing module, thus improving link prediction accuracy and insight.

**Strengths:**

1. The proposed method is technically sound in general.
2. The proposed approach could outperform baselines in general, and it has a strong performance in the cold-start scenarios.
3. The presentation is clear in general.

**Weaknesses:**

1. The novelty of the proposed method is limited, it's more like a mixture of many existing studies.
2. Many components of the Network-View representation learning part are designed for solving cold-start problems, how do these components impact models' performance in cold-start scenarios?
3. I'm not fully convinced by the contribution regarding the interpretability part, which is basically an attention map between entities. Any GNN with an attention mechanism can plot a similar interpretable attention map.

**Questions:**

Please see weaknesses.

---

### Note · Authors · 2024-11-26

I have read and agree with the venue's withdrawal policy on behalf of myself and my co-authors.